# Learning Calibratable Policies using Programmatic Style-Consistency

## Abstract

We study the important and challenging problem of controllable generation of long-term sequential behaviors. Solutions to this problem would impact many applications, such as calibrating behaviors of AI agents in games or predicting player trajectories in sports. In contrast to the well-studied areas of controllable generation of images, text, and speech, there are significant challenges that are unique to or exacerbated by generating long-term behaviors: how should we specify the factors of variation to control, and how can we ensure that the generated temporal behavior faithfully demonstrates diverse styles? In this paper, we leverage large amounts of raw behavioral data to learn policies that can be calibrated to generate a diverse range of behavior styles (e.g., aggressive versus passive play in sports). Inspired by recent work on leveraging programmatic labeling functions, we present a novel framework that combines imitation learning with data programming to learn style-calibratable policies. Our primary technical contribution is a formal notion of style-consistency as a learning objective, and its integration with conventional imitation learning approaches. We evaluate our framework using demonstrations from professional basketball players and agents in the MuJoCo physics environment, and show that our learned policies can be accurately calibrated to generate interesting behavior styles in both domains.

## 1 Introduction

The widespread availability of recorded tracking data is enabling the study of complex behaviors in many domains, including sports (Chen et al., 2016a; Le et al., 2017; Zhan et al., 2019), video games (Kurin et al., 2017; Broll et al., 2019), laboratory animals (Eyjolfsdottir et al., 2014; 2017; Johnson et al., 2016), facial expressions (Suwajanakorn et al., 2017; Taylor et al., 2017), commonplace activities such as cooking (Nishimura et al., 2019), and driving (Bojarski et al., 2016; Chang et al., 2019). The tracking data is often obtained from multiple experts and can exhibit very diverse styles (e.g., aggressive versus passive play in sports). Our work is motivated by the opportunity to maximally leverage these datasets by cleanly extracting such styles in addition to modeling the raw behaviors.

Our goal is to train policies that can be controlled, or calibrated, to produce different behavioral styles inherent in the demonstration data. For example, Figure 1a depicts demonstrations from real basketball players with variations of many types, including movement speed, desired destinations, tendencies for long versus short passes, and curvature of movement routes, amongst many others. A calibratable policy would be able to generate trajectories consistent with various styles, such as low movement speed as in Figure 1b, or approach the basket as in Figure 1c, or to both styles simultaneously as in Figure 1d. Importantly, we aim to train a single policy that can generate behaviors calibrated across multiple styles. Having such policies would empower many downstream tasks, including behavior discovery (Eyjolfsdottir et al., 2014), realistic simulations (Le et al., 2017), virtual agent design (Broll et al., 2019), and counterfactual behavioral reasoning (Zhan et al., 2019).

We focus on three research questions. The first question is strategic: what systematic form of domain knowledge can we leverage to quickly and cleanly extract style information from raw behavioral data? The second question is formulaic: how can we formalize the learning objective to encourage learning style-calibratable policies? The third question is algorithmic: how do we design practical learning approaches that reliably optimize the learning objective?

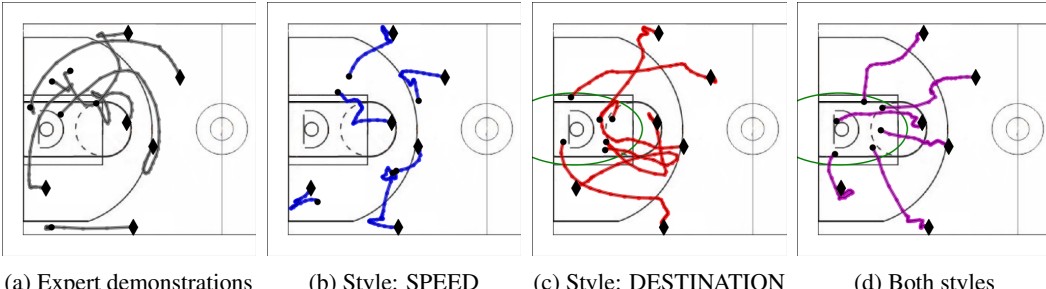

(a) Expert demonstrations   (b) Style: SPEED   (c) Style: DESTINATION   (d) Both styles

Figure 1: Basketball trajectories from policies that are: (a) the expert; (b) calibrated to move at low speeds; (c) calibrated to terminate near the basket (within green boundary); and (d) calibrated for both (b) & (c) simultaneously. Diamonds (♦) and dots (●) indicate initial and final positions.

To address these challenges, we present a novel framework inspired by *data programming* (Ratner et al., 2016), a paradigm in weak supervision that utilizes automated labeling procedures, called labeling functions, to learn without ground-truth labels. In our setting, labeling functions enable domain experts to quickly translate domain knowledge of diverse styles into programmatically generated style annotations. For instance, it is trivial to write programmatic labeling functions for the two styles—speed and destination—depicted in Figure 1. Labeling functions also motivate a metric for learning, which we call *programmatic style-consistency*, to evaluate calibration of policies: rollouts generated for a specific style should return the same style label when fed to the labeling function. Finally, our framework is generic and is easily integrated into conventional imitation learning approaches. To summarize, our contributions are:

- We propose a novel framework for learning policies calibrated to diverse behavior styles.
- Our framework allows users to express styles as labeling functions, which can be quickly applied to programmatically produce a weak signal of style labels.
- Our framework introduces *style-consistency* as a metric to evaluate calibration to styles.
- We present an algorithm to learn calibratable policies that maximize style-consistency of the generated behaviors, and validate it in basketball and simulated physics environments.

## 2    BACKGROUND: IMITATION LEARNING USING TRAJECTORY VAEs

Since our focus is on learning style-calibratable generative policies, for simplicity we develop our approach with the basic imitation learning paradigm of behavioral cloning using trajectory variational autoencoders, which we describe here. Interesting future directions include composing our approach with more advanced imitation learning approaches as well as with reinforcement learning.

**Notation.** Let $\mathcal{S}$ and $\mathcal{A}$ denote the environment state and action spaces. At each timestep $t$, an agent observes state $\mathbf{s}_t \in \mathcal{S}$ and executes action $\mathbf{a}_t \in \mathcal{A}$ using a policy $\pi : \mathcal{S} \to \mathcal{A}$. The environment then transitions to the next state $\mathbf{s}_{t+1}$ according to a (typically unknown) dynamics function $f : \mathcal{S} \times \mathcal{A} \to \mathcal{S}$. For the rest of this paper, we assume $f$ is deterministic; a modification of our approach for stochastic $f$ is included in Appendix B. A trajectory $\tau$ is a sequence of $T$ state-action pairs and the last state: $\tau = \{(\mathbf{s}_t, \mathbf{a}_t)\}_{t=1}^T \cup \{\mathbf{s}_{T+1}\}$. Let $\mathcal{D}$ be a set of $N$ trajectories collected from expert demonstrations. In our experiments, each trajectory in $\mathcal{D}$ has the same length $T$, but in general this does not need to be the case.

**Learning objective.** We begin with the basic imitation learning paradigm of behavioral cloning (Syed & Schapire, 2008). The goal is to learn a policy that behaves like the pre-collected demonstrations:

$$\pi^* = \arg\min_{\pi} \mathbb{E}_{\tau \sim \mathcal{D}} \left[ \mathcal{L}^{\text{imitation}}(\tau, \pi) \right], \tag{1}$$

where $\mathcal{L}^{\text{imitation}}$ is a loss function that quantifies the mismatch between the actions chosen by $\pi$ and those in the demonstrations. Since we are primarily interested in probabilistic or generative policies, we typically use (variations of) negative log-likelihood: $\mathcal{L}(\tau, \pi) = \sum_{t=1}^T -\log \pi(\mathbf{a}_t|\mathbf{s}_t)$, where $\pi(\mathbf{a}_t|\mathbf{s}_t)$ is the probability of $\pi$ choosing action $\mathbf{a}_t$ in state $\mathbf{s}_t$.

**Trajectory Variational Autoencoders.** A common model choice for instantiating $\pi$ is the trajectory variational autoencoder (TVAE), which is a sequential generative model built on top of variational autoencoders (Kingma & Welling, 2014), and have been shown to work well in a range of generative policy learning settings (Wang et al., 2017; Ha & Eck, 2018; Co-Reyes et al., 2018). In its simplest form, a TVAE introduces a latent variable $\mathbf{z}$ (also called a trajectory embedding) with prior distribution $p$, an encoder network $q_\phi$, and a policy decoder $\pi_\theta$. Its imitation learning objective is:

$$\mathcal{L}^{\text{tvae}}(\tau, \pi_\theta; q_\phi) = \mathbb{E}_{q_\phi(\mathbf{z}|\tau)} \left[ \sum_{t=1}^{T} - \log \pi_\theta(\mathbf{a}_t|\mathbf{s}_t, \mathbf{z}) \right] + D_{KL}\big(q_\theta(\mathbf{z}|\tau)||p(\mathbf{z})\big). \tag{2}$$

The main shortcoming of TVAEs and related approaches, which we address in Sections 3 & 4, is that the resulting policies cannot be easily calibrated to generate specific styles of behavior. For instance, the goal of the trajectory embedding $\mathbf{z}$ is to capture all the styles that exist in the expert demonstrations, but there is no guarantee that the embeddings cleanly encode the desired styles in a calibrated way. Previous work has largely relied on unsupervised learning techniques that either require significant domain knowledge (Le et al., 2017), or have trouble scaling to complex styles commonly found in real-world applications (Wang et al., 2017; Li et al., 2017).

## 3 PROGRAMMATIC STYLE-CONSISTENCY

Building upon the basic setup in Section 2, we focus on the setting where the demonstrations $\mathcal{D}$ contain diverse behavior styles. To start, let $\mathbf{y} \in Y$ denote a single style label (e.g., speed or destination, as shown in Figure 1). Our goal is to learn a policy $\pi$ that can be explicitly calibrated to $\mathbf{y}$, i.e., trajectories generated by $\pi(\cdot|\mathbf{y})$ should match the demonstrations in $\mathcal{D}$ that exhibit style $\mathbf{y}$.

Obtaining style labels can be expensive using conventional annotation methods, and unreliable using unsupervised approaches. We instead utilize easily programmable labeling functions that automatically produce style labels, described next. We then formalize a notion of style-consistency as a learning objective, and in Section 4 describe a practical learning approach.

**Labeling functions.** Introduced in the data programming paradigm (Ratner et al., 2016), labeling functions programmatically produce weak and noisy labels to learn models on otherwise unlabeled datasets. A significant benefit is that labeling functions are often simple scripts that can be quickly applied to the dataset, which is much cheaper than manual annotations and more reliable than unsupervised methods. In our framework, we study behavior styles that can be represented as labeling functions, which we denote $\lambda$, that map trajectories $\tau$ to style labels $\mathbf{y}$. A simple example is:

$$\lambda(\tau) = \mathbb{1}\{\|\mathbf{s}_{T+1} - \mathbf{s}_1\|_2 > c\}, \tag{3}$$

which distinguishes between trajectories with large (greater than a threshold $c$) versus small total displacement. We experiment with a range of labeling functions, as described in Section 6. Multiple labeling functions can be provided at once, possibly from multiple users. Many behavior styles used in previous work can be represented as labeling functions, e.g., agent speed (Wang et al., 2017). We use trajectory-level labels $\lambda(\tau)$ in our experiments, but in general labeling functions can be applied on subsequences $\lambda(\tau_{t:t+h})$ to obtain per-timestep labels. We can efficiently annotate datasets using labeling functions, which we denote as $\lambda(\mathcal{D}) = \{(\tau_i, \lambda(\tau_i))\}_{i=1}^{N}$. Our goal can now be phrased as: given $\lambda(\mathcal{D})$, train a policy $\pi : \mathcal{S} \times Y \mapsto \mathcal{A}$ such that $\pi(\cdot|\mathbf{y})$ is calibrated to styles $\mathbf{y}$ found in $\lambda(\mathcal{D})$.

**Style-consistency.** A key insight in our work is that labeling functions naturally induce a metric for calibration. If a policy $\pi(\cdot|\mathbf{y})$ is calibrated to $\lambda$, we would expect the generated behaviors to be consistent with the label. So, we expect the following loss to be small:

$$\mathbb{E}_{\mathbf{y}\sim p(\mathbf{y}), \tau\sim\pi(\cdot|\mathbf{y})} \left[ \mathcal{L}^{\text{style}}\big(\lambda(\tau), \mathbf{y}\big) \right], \tag{4}$$

where $p(\mathbf{y})$ is a prior over the style labels, and $\tau$ is obtained by executing the style-conditioned policy in the environment. $\mathcal{L}^{\text{style}}$ is thus a disagreement loss over labels that is minimized at $\lambda(\tau) = \mathbf{y}$, e.g., $\mathcal{L}^{\text{style}}\big(\lambda(\tau), \mathbf{y}\big) = \mathbb{1}\{\lambda(\tau) \neq \mathbf{y}\}$ for categorical labels. We refer to (4) as the *style-consistency* loss, and say that $\pi(\cdot|\mathbf{y})$ is maximally calibrated to $\lambda$ when (4) is minimized. Our full learning objective incorporating (4) with (1) is:

$$\pi^* = \arg\min_{\pi} \mathbb{E}_{(\tau, \lambda(\tau))\sim\lambda(\mathcal{D})} \left[ \mathcal{L}^{\text{imitation}}\Big(\tau, \pi\big(\cdot \mid \lambda(\tau)\big)\Big) \right] + \mathbb{E}_{\mathbf{y}\sim p(\mathbf{y}), \tau\sim\pi(\cdot|\mathbf{y})} \left[ \mathcal{L}^{\text{style}}\big(\lambda(\tau), \mathbf{y}\big) \right]. \tag{5}$$

The simplest choice for the prior distribution $p(\mathbf{y})$ is the marginal distribution of styles in $\lambda(\mathcal{D})$. The first term in (5) is a standard imitation learning objective and can be tractably estimated using $\lambda(\mathcal{D})$. To enforce style-consistency with the second term, conceptually we need to sample several $\mathbf{y} \sim p(\mathbf{y})$, then several rollouts $\tau \sim \pi(\cdot \mid \mathbf{y})$ from the current policy, and query the labeling function for each of them. Furthermore, if $\lambda$ is a non-differentiable function defined over the entire trajectory, as is the case in (3), then we cannot simply backpropagate the style-consistency loss. In Section 4, we introduce differentiable approximations to more easily optimize the challenging objective in (5).

**Multiple styles.** Our notion of style-consistency can be easily extended to simultaneously optimize for multiple styles. Suppose we have $M$ labeling functions $\{\lambda_i\}_{i=1}^M$ and corresponding label spaces $\{Y_i\}_{i=1}^M$. Let $\lambda$ denote $(\lambda_1, \ldots, \lambda_M)$ and $\mathbf{y}$ denote $(\mathbf{y}_1, \ldots, \mathbf{y}_M)$. Then style-consistency becomes:

$$\mathbb{E}_{\mathbf{y} \sim p(\mathbf{y}), \tau \sim \pi(\cdot | \mathbf{y})} \left[ \sum_{i=1}^M \mathcal{L}_i^{\text{style}} \big( \lambda_i(\tau), \mathbf{y}_i \big) \right]. \tag{6}$$

Note that style-consistency is optimized when the generated trajectory agrees with *all* labeling functions. Although this can be very challenging to achieve, it describes the most desirable outcome, i.e. $\pi(\cdot | \mathbf{y})$ is a policy that can be calibrated to *all* styles simultaneously.

## 4 Learning Approach

Optimizing (5) is challenging due to the long-time horizon and non-differentiability of the labeling functions $\lambda$.[1] Given unlimited queries to the environment, one could naively employ model-free reinforcement learning, e.g., estimating (4) using rollouts and optimizing using policy gradient ap-

---
**Algorithm 1** Generic recipe for optimizing (5)
---
1: **Input**: demonstrations $\mathcal{D}$, labeling functions $\lambda$
2: construct $\lambda(\mathcal{D})$ by applying $\lambda$ on trajectories in $\mathcal{D}$
3: optimize (7) to convergence to learn $C_{\psi^*}^\lambda$
4: optimize (8) to convergence to learn $\pi^*$
---

proaches. We instead take a model-based approach, described generically in Algorithm 1, that is more computationally-efficient and decomposable. The advantages of our approach are that it is compatible with batch or offline learning, and enables easier diagnosis of deficiencies in the algorithmic framework. To develop our approach, we first introduce a label approximator for $\lambda$, and then show how to optimize through the environmental dynamics using a differentiable model-based learning approach.

**Approximating labeling functions.** To deal with non-differentiability of $\lambda$, we approximate it with a differentiable function $C_\psi^\lambda$ parameterized by $\psi$:

$$\psi^* = \arg\min_\psi \mathbb{E}_{(\tau, \lambda(\tau)) \sim \lambda(\mathcal{D})} \left[ \mathcal{L}^{\text{label}} \big( C_\psi^\lambda(\tau), \lambda(\tau) \big) \right]. \tag{7}$$

Here, $\mathcal{L}^{\text{label}}$ is a differentiable loss that approximates $\mathcal{L}^{\text{style}}$, such as cross-entropy loss when $\mathcal{L}^{\text{style}}$ is the $0/1$ loss. In our experiments we use a recurrent neural net to represent $C_\psi^\lambda$. We then modify the style-consistency term in (5) with $C_{\psi^*}^\lambda$ and optimize:

$$\pi^* = \arg\min_\pi \mathbb{E}_{(\tau, \lambda(\tau)) \sim \lambda(\mathcal{D})} \left[ \mathcal{L}^{\text{imitation}} \big( \tau, \pi(\cdot \mid \lambda(\tau)) \big) \right] + \mathbb{E}_{\mathbf{y} \sim p(\mathbf{y}), \tau \sim \pi(\cdot | \mathbf{y})} \left[ \mathcal{L}^{\text{label}} \big( C_{\psi^*}^\lambda(\tau), \mathbf{y} \big) \right]. \tag{8}$$

**Optimizing $\mathcal{L}^{\text{style}}$ over trajectories.** The next challenge to be addressed is one of credit assignment over time steps. For instance, consider the labeling function in (3) that computes the difference between the first and last states. Our label approximator $C_{\psi^*}^\lambda$ may converge to a solution that ignores all inputs except for $\mathbf{s}_1$ and $\mathbf{s}_{T+1}$. In this case, gradient descent through $C_{\psi^*}^\lambda$ provides no information about intermediate timesteps. In other words, effective optimization of style-consistency in (8) requires informative learning signals on all actions taken by the policy.

In general, there are two types of approaches to address this challenge: model-free and model-based. A model-free solution views this credit assignment challenge as analogous to that faced by RL, and

---

[1]This issue is not encountered in previous work on style-dependent imitation learning (Li et al., 2017; Hausman et al., 2017), since they use purely unsupervised methods such as maximizing mutual information.

repurposes generic reinforcement learning algorithms. We instead choose a model-based approach for two reasons: (a) we found it to be compositionally simpler and easier to debug; and (b) we can use the learned model to obtain hallucinated rollouts of the current policy efficiently during training.

**Modeling dynamics for credit assignment.** Our model-based approach utilizes a dynamics model $M_\varphi$ to approximate the environment's dynamics by predicting the change in state given the current state and action:

$$\varphi^* = \arg\min_\varphi \mathbb{E}_{\tau \sim \mathcal{D}} \sum_{t=1}^{T} \mathcal{L}^{\text{dynamics}}\big(M_\varphi(\mathbf{s}_t, \mathbf{a}_t), (\mathbf{s}_{t+1} - \mathbf{s}_t)\big), \tag{9}$$

where $\mathcal{L}^{\text{dynamics}}$ is often $L_2$ or squared-$L_2$ loss (Nagabandi et al., 2018; Luo et al., 2019). This allows us to generate trajectories by rolling out: $\mathbf{s}_{t+1} = \mathbf{s}_t + M_\varphi\big(\mathbf{s}_t, \pi(\mathbf{s}_t)\big)$. Then optimizing for style-consistency in (8) would backpropagate through our dynamics model $M_\varphi$ and provide informative learning signals to the policy at every timestep.

We outline our model-based approach in Algorithm 2. Lines 10-12 describe an optional step to fine-tune the dynamics model by querying the environment for trajectories of the current policy (similar to Luo et al. (2019)); we found that this can help improve style-consistency in some experiments.

---

**Algorithm 2** Model-based approach for optimizing style-consistency

---

1: **Input**: demonstrations $\mathcal{D}$, labeling function $\lambda$, label approximator $C_\psi^\lambda$, dynamics model $M_\varphi$
2: $\lambda(\mathcal{D}) \leftarrow \big\{\big(\tau_i, \lambda(\tau_i)\big)\big\}_{i=1}^{N}$
3: **for** $n_{\text{dynamics}}$ iterations **do**
4:     optimize (9) with batch from $\mathcal{D}$             ▷ Train dynamics model $M_\varphi$
5: **for** $n_{\text{label}}$ iterations **do**
6:     optimize (7) with batch from $\lambda(\mathcal{D})$         ▷ Train label approximator $C_\psi^\lambda$
7: **for** $n_{\text{policy}}$ iterations **do**
8:     $\mathcal{B} \leftarrow \{$ collect $n_{\text{collect}}$ trajectories with $M_\varphi$ and current policy $\pi$ $\}$
9:     optimize (8) with batch from $\lambda(\mathcal{D})$ and $\mathcal{B}$         ▷ Train policy $\pi$
10:     **for** $n_{\text{env}}$ iterations **do**
11:         $\tau_{\text{env}} \leftarrow$ collect 1 trajectory from environment with $\pi$
12:         optimize (9) with $\tau_{\text{env}}$         ▷ Fine-tune dynamics model $M_\varphi$

---

## 5 RELATED WORK

Our work combines ideas from imitation learning and data programming, developing a weakly supervised approach for more explicit and fine-grained calibration. This is related to learning disentangled representations and controllable generative modeling, reviewed below.

**Imitation learning of diverse behaviors** has focused on unsupervised approaches to infer latent variables/codes that capture behavior styles (Li et al., 2017; Hausman et al., 2017; Wang et al., 2017). Similar approaches have also been studied for generating text conditioned on attributes such as sentiment or tense (Hu et al., 2017). A typical strategy is to maximize the mutual information between the latent codes and trajectories, in contrast to our notion of programmatic style-consistency.

**Disentangled representation learning** aims to learn representations where each latent dimension corresponds to exactly one desired factor of variation (Bengio et al., 2012). Recent studies (Locatello et al., 2019) have noted that popular techniques (Chen et al., 2016b; Higgins et al., 2017; Kim & Mnih, 2018; Chen et al., 2018) can be sensitive to hyperparameters and that evaluation metrics can be correlated with certain model classes and datasets, which suggests that unsupervised learning approaches may, in general, be unreliable for discovering cleanly calibratable representations.

**Conditional generation** for images has recently focused on *attribute manipulation* (Bao et al., 2017; Creswell et al., 2017; Klys et al., 2018), which aims to enforce that changing a label affects only one aspect of the image while keeping everything else the same (similar to disentangled representation learning). We extend these models and compare with our approach in Section 6. Our experimental results suggest that these algorithms do not necessarily scale well into sequential domains.

**Enforcing consistency in generative modeling**, such as cycle-consistency in image generation (Zhu et al., 2017), and self-consistency in hierarchical reinforcement learning (Co-Reyes et al.,

2018) has proved beneficial. The former minimizes a discriminative disagreement, whereas the latter minimizes a distributional disagreement between two sets of generated behaviors (e.g., KL-divergence). From this perspective, our style-consistency notion is more similar to the former; however we also enforce consistency over multiple time-steps, which is more similar to the latter.

## 6 EXPERIMENTS

We first briefly describe our experimental setup and choice of baselines, and then discuss our main experimental results. A full description of the experiments is available in Appendix C.

**Data.** We validate our framework on two datasets: 1) a collection of professional basketball player trajectories with the goal of learning a policy that generates realistic player-movement, and 2) a Cheetah agent running horizontally in MuJoCo (Todorov et al., 2012) with the goal of learning a policy with calibrated gaits. The former has a known dynamics function: $f(\mathbf{s}_t, \mathbf{a}_t) = \mathbf{s}_t + \mathbf{a}_t$, where $\mathbf{s}_t$ and $\mathbf{a}_t$ are the player's position and velocity on the court respectively; we expect the dynamics model $M_\varphi$ to easily recover this function. The latter has an unknown dynamics function (which we learn a model of when approximating style-consistency). We obtain Cheetah demonstrations from a collection of policies trained using `pytorch-a2c-ppo-acktr` (Kostrikov, 2018) to interface with the DeepMind Control Suite's Cheetah domain (Tassa et al., 2018)—see Appendix C for details.

**Labeling functions.** Labeling functions for Basketball include: 1) average SPEED of the player, 2) DISPLACEMENT from initial to final position, 3) distance from final position to a fixed DESTINATION on the court (e.g. the basket), 4) mean DIRECTION of travel, and 5) CURVATURE of the trajectory, which measures the player's propensity to change directions. For Cheetah, we have labeling functions for the agent's 1) SPEED, 2) TORSO HEIGHT, 3) BACK-FOOT HEIGHT, and 4) FRONT-FOOT HEIGHT that can be trivially extracted from the environment.

We threshold the aforementioned labeling functions into categorical labels (leaving real-valued labels for future work) and use (4) for style-consistency with $\mathcal{L}^{\text{style}}$ as the $0/1$ loss. We use cross-entropy for $\mathcal{L}^{\text{label}}$ and list all other hyperparameters in Appendix C. Whenever we report style-consistency results, we use $1 - \mathcal{L}^{\text{style}}$ in (4) so that all results are easily interpreted as accuracies.

**Baselines.** We compare our approach, CTVAE-style, with 3 baseline policy models:

1. **CTVAE**: The conditional version of TVAEs (Wang et al., 2017).
2. **CTVAE-info**: CTVAE with information factorization (Creswell et al., 2017) that *implicitly* maximizes style-consistency by removing all information correlated with **y** from **z**.
3. **CTVAE-mi**: CTVAE with mutual information maximization between style labels and trajectories. This is a supervised variant of unsupervised models (Chen et al., 2016b; Li et al., 2017), and also requires learning a dynamics model for sampling policy rollouts.

Detailed descriptions and model parameters of baselines are in Appendix A and C respectively. All models build upon TVAEs, which are also conditioned on a latent variable (see Section 2). We highlight that the underlying model choice is orthogonal to our contributions; our framework is compatible with any imitation learning algorithm (see Table 13 in Appendix).

### 6.1 HOW WELL CAN WE CALIBRATE POLICIES FOR INDIVIDUAL STYLES?

We first threshold labeling functions into 3 classes for Basketball and 2 classes for Cheetah; the marginal distribution $p(\mathbf{y})$ of styles in $\lambda(\mathcal{D})$ is roughly uniform over these classes. Then we learn a policy $\pi^*$ calibrated to each of these styles. Finally, we generate rollouts from each of the learned policies to measure style-consistency. Table 1 compares the median style-consistency (over 5 seeds) of learned policies. For Basketball, CTVAE-style significantly outperforms baselines and achieves almost perfect style-consistency for 4 of the 5 styles (the best style-consistency over 5 seeds outperforms *all* baselines, shown in Tables 8a and 9a in Appendix C). For Cheetah, CTVAE-style outperforms all baselines, but the absolute performance is lower than for Basketball (mostly due to the more complex environment dynamics).

We visualize our CTVAE-style policy calibrated for DESTINATION(net) (with style-consistency of 0.97) in Figure 2. The green boundaries divide the court into 3 regions, one for each label class. Policy rollouts almost always terminate in the corresponding region of the label class. Note that

| | **Basketball** | | | | | **Cheetah** | | | |
|---|---|---|---|---|---|---|---|---|---|
| **Model** | **Speed** | **Disp.** | **Dest.** | **Dir.** | **Curve** | **Speed** | **Torso** | **BFoot** | **FFoot** |
| CTVAE | 83 | 72 | 82 | 77 | 61 | 59 | 63 | 68 | 68 |
| CTVAE-info | 84 | 71 | 79 | 72 | 60 | 57 | 63 | 65 | 66 |
| CTVAE-mi | 86 | 74 | 82 | 77 | **72** | 60 | 65 | 65 | 70 |
| CTVAE-style | **95** | **96** | **97** | **97** | 68 | **79** | **80** | **80** | **77** |

Table 1: **Individual Style Calibration:** Style-consistency ($\times 10^{-2}$, median over 5 seeds) of policies evaluated with 4,000 Basketball and 500 Cheetah rollouts. Trained separately for each style, CTVAE-style policies outperform baselines for all styles in Cheetah and 4/5 styles in Basketball.

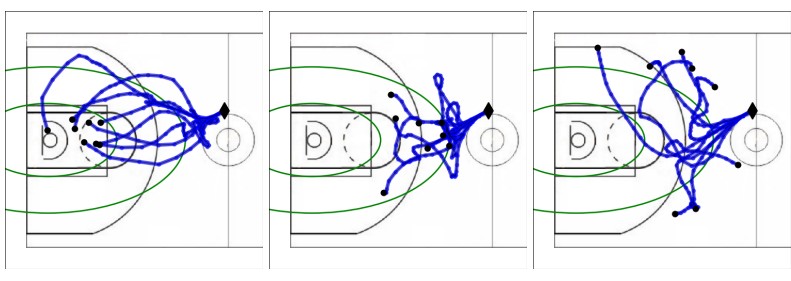

(a) Label class 0 (close)     (b) Label class 1 (mid)     (c) Label class 2 (far)

Figure 2: CTVAE-style rollouts calibrated for `DESTINATION(net)`, 0.97 style-consistency. Diamonds (♦) and dots (•) indicate initial and final positions. Regions divided by green lines represent label classes.

although the policy is calibrated for one style, rollouts still exhibit diverse behaviors (i.e. distribution of trajectories did not collapse into a single mode), which suggests that there are other styles being imitated. Section 6.2 examines this further by testing calibration to multiple styles simultaneously.

We also consider cases in which labeling functions can have several classes and non-uniform distributions (i.e. some styles are more/less common than others). We threshold `DESTINATION(net)` into 6 classes for Basketball and `SPEED` into 4 classes for Cheetah and compare the policies in Table 2. In general, we observe degradation in overall style-consistency accuracies as the number of classes increase. However, CTVAE-style policies still consistently achieve better style-consistency than baselines in this setting as well. In the appendix, we visualize all 6 classes of `DESTINATION(net)` in Figure 4 and include another experiment with up to 8 classes of `DISPLACEMENT` in Table 8c. These results suggest that incorporating programmatic style-consistency while training via (8) can yield good qualitative and quantitative calibration results.

| | **Basketball** - `DESTINATION(net)` | | | | **Cheetah** - `SPEED` | |
|---|---|---|---|---|---|---|
| **Model** | **2 classes** | **3 classes** | **4 classes** | **6 classes** | **3 classes** | **4 classes** |
| CTVAE | 87 | 82 | 78 | 74 | 45 | 37 |
| CTVAE-info | 87 | 81 | 75 | 77 | 49 | 39 |
| CTVAE-mi | 88 | 81 | 74 | 76 | 48 | 37 |
| CTVAE-style | **98** | **97** | **89** | **84** | **59** | **51** |

Table 2: **Fine-grained Style-consistency:** ($\times 10^{-2}$, median over 5 seeds) Training on labeling functions with more classes yields increasingly fine-grained calibration of behavior. Although CTVAE-style degrades as the number of classes increases, it outperforms baselines for all styles.

## 6.2 CAN WE CALIBRATE POLICIES FOR MULTIPLE STYLES SIMULTANEOUSLY?

We now consider multiple style-consistency as in (6), which measures the total accuracy with *all* labeling functions simultaneously. For instance, in addition to terminating close to the net in Figure 2, a user may also want to control the speed at which the agent moves towards the target destination.

Table 3 compares the style-consistency of policies calibrated for up to 5 styles for Basketball and 3 styles for Cheetah. Calibrating for multiple styles simultaneously is a very difficult task for baselines, as their style-consistency degrades significantly as the number of styles increases. On the other hand, CTVAE-style sees a modest decrease in style-consistency but is still significantly better calibrated (0.75 style-consistency for *all* 5 styles vs. only 0.30 for the best baseline in Basketball).

| Model | Basketball | | | | Cheetah | |
|---|---|---|---|---|---|---|
| | 2 styles | 3 styles | 4 styles | 5 styles | 2 styles | 3 styles |
| CTVAE | 71 | 58 | 50 | 37 | 41 | 28 |
| CTVAE-info | 69 | 58 | 51 | 32 | 41 | 27 |
| CTVAE-mi | 72 | 56 | 51 | 30 | 40 | 28 |
| CTVAE-style | **93** | **88** | **88** | **75** | **54** | **40** |

Table 3: **Multi Style-consistency:** ($10^{-2}$, median over 5 seeds) Simultaneously calibrated to multiple styles, CTVAE-style policies outperform baselines for all styles in Cheetah and in Basketball.

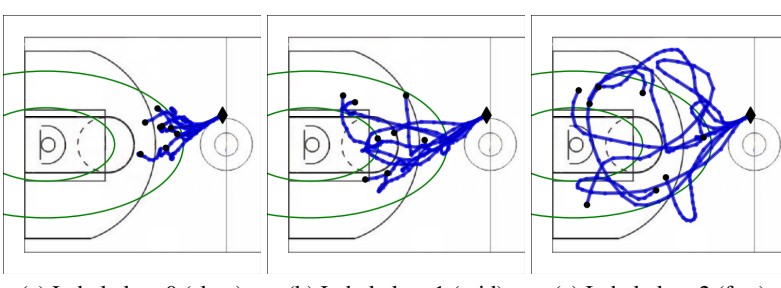

(a) Label class 0 (slow)    (b) Label class 1 (mid)    (c) Label class 2 (fast)

Figure 3: CTVAE-style rollouts calibrated for 2 styles: label class 1 of `DESTINATION(net)` (see Figure 2) and each class for `SPEED`, with 0.93 style-consistency. Diamonds (♦) and dots (●) indicate initial and final positions.

We visualize a CTVAE-style policy calibrated for two styles in Basketball with style-consistency 0.93 in Figure 3. CTVAE-style outperforms baselines in Cheetah as well, but there is still room for improvement to reach maximal style-consistency in future work.

### 6.3 WHAT IS THE TRADE-OFF BETWEEN STYLE-CONSISTENCY AND IMITATION QUALITY?

In Table 4, we investigate whether CTVAE-style's superior style-consistency is attained at a significant cost to imitation quality, since we jointly optimize both in (5). For Basketball, high style-consistency is achieved without any degradation in imitation quality. For Cheetah, negative log-likelihood is slightly worse; a followup experiment in Table 12 of the appendix shows that we can improve imitation quality with further training, which can sometimes modestly decrease style-consistency.

| Model | Basketball | | Cheetah | |
|---|---|---|---|---|
| | $D_{KL}$ | NLD | $D_{KL}$ | NLD |
| TVAE | 2.5 | -7.9 | 29 | -0.60 |
| CTVAE | 2.5 | -8.0 | 29 | -0.59 |
| CTVAE-info | 2.3 | -7.9 | 29 | -0.58 |
| CTVAE-mi | 2.6 | -8.0 | 29 | -0.57 |
| CTVAE-style | 2.3 | -7.8 | 30 | -0.28 |

Table 4: KL-divergence and negative log-density per timestep for TVAE models (lower is better). CTVAE-style is comparable to baselines for Basketball, but is slightly worse for Cheetah.

## 7 CONCLUSION AND FUTURE WORK

We propose a novel framework for imitating diverse behavior styles while also calibrating to desired styles. Our framework leverages labeling functions to tractably represent styles and introduces programmatic style-consistency, a metric that allows for fair comparison between calibrated policies. Our experiments demonstrate strong empirical calibration results.

We believe that our framework lays the foundation for many directions of future research. First, can one model more complex styles not easily captured with a single labeling function (e.g. aggressive vs. passive play in sports) by composing simpler labeling functions (e.g. max speed, distance to closest opponent, number of fouls committed, etc.), similar to (Ratner et al., 2016; Bach et al., 2017)? Second, can we use these per-timestep labels to model transient styles, or simplify the credit assignment problem when learning to calibrate? Third, can we blend our programmatic supervision with unsupervised learning approaches to arrive at effective semi-supervised solutions? Fourth, can we use leverage model-free approaches to further optimize self-consistency, e.g., to fine-tune from our model-based approach? Finally, can we integrate our framework with reinforcement learning to also optimize for environmental rewards?

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

## A   BASELINE POLICY MODELS

**1) Conditional-TVAE (CTVAE).**   The conditional version of TVAEs optimizes:

$$\mathcal{L}^{\text{ctvae}}(\tau, \pi_\theta; q_\phi) = \mathbb{E}_{q_\phi(\mathbf{z}|\tau,\mathbf{y})} \left[ \sum_{t=1}^{T} -\log \pi_\theta(\mathbf{a}_t|\mathbf{s}_t, \mathbf{z}, \mathbf{y}) \right] + D_{KL}\big(q_\theta(\mathbf{z}|\tau,\mathbf{y})||p(\mathbf{z})\big). \qquad (10)$$

**2) CTVAE with information factorization (CTVAE-info).**   (Creswell et al., 2017; Klys et al., 2018) augment conditional-VAE models with an auxiliary network $A_\psi(\mathbf{z})$ which is trained to predict the label $\mathbf{y}$ from $\mathbf{z}$, while the encoder $q_\phi$ is also trained to minimize the accuracy of $A_\psi$. This model *implicitly* maximizes self-consistency by removing the information correlated with $\mathbf{y}$ from $\mathbf{z}$, so that any information pertaining to $\mathbf{y}$ that the decoder needs for reconstruction must all come from $\mathbf{y}$. While this model was previously used for image generation, we extend it into the sequential domain:

$$\max_{\theta,\phi} \left( \mathbb{E}_{q_\phi(\mathbf{z}|\tau)} \left[ \min_\psi \mathcal{L}^{\text{aux}}\big(A_\psi(\mathbf{z}), \mathbf{y}\big) + \sum_{t=1}^{T} \log \pi_\theta(\mathbf{a}_t|\mathbf{s}_t, \mathbf{z}, \mathbf{y}) \right] - D_{KL}\big(q_\theta(\mathbf{z}|\tau)||p(\mathbf{z})\big) \right). \quad (11)$$

Note that the encoder in (10) and (11) differ in that $q_\phi(\mathbf{z}|\tau)$ is no longer conditioned on the label $\mathbf{y}$.

**3) CTVAE with mutual information maximization (CTVAE-mi).**   In addition to (10), we can also maximize the mutual information between labels and trajectories $I(\mathbf{y};\tau)$. This quantity is hard to maximize directly, so instead we maximize the variational lower bound:

$$I(\mathbf{y};\tau) \geq \mathbb{E}_{\mathbf{y}\sim p(\mathbf{y}),\tau\sim\pi_\theta(\cdot|\mathbf{z},\mathbf{y})} \big[ \log r_\psi(\mathbf{y}|\tau) \big] + \mathcal{H}(\mathbf{y}), \qquad (12)$$

where $r_\psi$ approximates the true posterior $p(\mathbf{y}|\tau)$. In our setting, the prior over labels is known, so $\mathcal{H}(\mathbf{y})$ is a constant. Thus, the learning objective is:

$$\mathcal{L}^{\text{ctvae-mi}}(\tau, \pi_\theta; q_\phi) = \mathcal{L}^{\text{ctvae}}(\tau, \pi_\theta) + \mathbb{E}_{\mathbf{y}\sim p(\mathbf{y}),\tau\sim\pi_\theta(\cdot|\mathbf{z},\mathbf{y})} \big[ -\log r_\psi(\mathbf{y}|\tau) \big]. \qquad (13)$$

Optimizing (13) also requires collecting rollouts with the current policy, so similarly we also pretrain and fine-tune a dynamics model $M_\varphi$. This baseline can be interpreted as a supervised analogue of unsupervised models that maximize mutual information in (Li et al., 2017; Hausman et al., 2017).

## B   STOCHASTIC DYNAMICS FUNCTION

If the dynamics function $f$ of the environment is stochastic, we modify our approach in Algorithm 2 by changing the form of our dynamics model. We can model the change in state as a Gaussian distribution and minimize the negative log-likelihood:

$$\varphi_\mu^*, \varphi_\sigma^* = \arg\min_{\varphi_\mu,\varphi_\mu} \mathbb{E}_{\tau\sim\mathcal{D}} \sum_{t=1}^{T} -\log p(\Delta_t; \mu_t, \sigma_t), \qquad (14)$$

where $\Delta_t = \mathbf{s}_{t+1} - \mathbf{s}_t$, $\mu_t = M_{\varphi_\mu}(\mathbf{s}_t, \mathbf{a}_t)$, $\sigma_t = M_{\varphi_\sigma}(\mathbf{s}_t, \mathbf{a}_t)$, and $M_{\varphi_\mu}$, $M_{\varphi_\sigma}$ are neural networks that can share weights. We can sample a change in state during rollouts using the reparametrization trick (Kingma & Welling, 2014), which allows us to backpropagate through the dynamics model during training.

## C   EXPERIMENT DETAILS

**Dataset details.**   See Table 5. Basketball trajectories are collected from tracking real players in the NBA. Figure 5 shows the distribution of basketball labeling functions applied on the training set. For Cheetah, we train 125 policies using PPO (Schulman et al., 2017) to run forwards at speeds ranging from 0 to 4 (m/s). We collect 25 trajectories per policy by sampling actions from the policy. We use (Kostrikov, 2018) to interface with (Tassa et al., 2018). Figure 6 shows the distributions of Cheetah labeling functions applied on the training set.

**Training hyperparameters.**   See Table 6.

**Model parameters.** We model all trajectory embeddings **z** as a diagonal Gaussian with a standard normal prior. Encoder $q_\phi$ and label approximators $C_\psi^\lambda$ are bi-directional GRUs (Cho et al., 2014) followed by linear layers. Policy $\pi_\theta$ is recurrent for basketball, but not for Cheetah. The Gaussian log sigma returned by $\pi_\theta$ is state-dependent for basketball, but state-independent for Cheetah. For Cheetah, we made these choices based on prior work in Mujoco for training gait policies. For Basketball, we observed a lot more variation in the 500k demonstrations so we experimented with more flexible model classes. See Table 7 for more model details.

| | $|\mathcal{S}|$ | $|\mathcal{A}|$ | $T$ | $N_{\text{train}}$ | $N_{\text{test}}$ | frequency (Hz) |
|---|---|---|---|---|---|---|
| Basketball | 2 | 2 | 24 | 520,015 | 67,320 | 3 |
| Cheetah | 18 | 6 | 200 | 2,500 | 625 | 40 |

Table 5: Dataset parameters for basketball and Cheetah environments.

| | batch size | # batch $b$ | $n_{\text{dynamics}}$ | $n_{\text{label}}$ | $n_{\text{policy}}$ | $n_{\text{collect}}$ | $n_{\text{env}}$ | learning rate |
|---|---|---|---|---|---|---|---|---|
| Basketball | 128 | 4,063 | $10 \cdot b$ | $20 \cdot b$ | $30 \cdot b$ | 128 | 0 | $2 \cdot 10^{-4}$ |
| Cheetah | 16 | 157 | $50 \cdot b$ | $20 \cdot b$ | $60 \cdot b$ | 16 | 1 | $10^{-3}$ |

Table 6: Hyperparameters for Algorithm 2. $b$ is the number of batches to see all trajectories in the dataset once. We also use $L_2$ regularization of $10^{-5}$ for training the dynamics model $M_\varphi$.

| | **z**-dim | $q_\phi$ GRU | $C_\psi^\lambda$ GRU | $\pi_\theta$ GRU | $\pi_\theta$ sizes | $M_\varphi$ sizes |
|---|---|---|---|---|---|---|
| Basketball | 4 | 128 | 128 | 128 | (128,128) | (128,128) |
| Cheetah | 8 | 200 | 200 | - | (200,200) | (500,500) |

Table 7: Model parameters for basketball and Cheetah environments.

| Model | Speed | | | Displacement | | | Destination | | | Direction | | | Curvature | | |
|---|---|---|---|---|---|---|---|---|---|---|---|---|---|---|---|
| CTVAE | 82 | 83 | 85 | 71 | 72 | 74 | 81 | 82 | 82 | 76 | 77 | 80 | 60 | 61 | 62 |
| CTVAE-info | **84** | 84 | 87 | 69 | 71 | 74 | 78 | 79 | 83 | 71 | 72 | 74 | **60** | 60 | 62 |
| CTVAE-mi | 84 | 86 | 87 | 71 | 74 | 74 | 80 | 82 | 84 | 75 | 77 | 78 | 58 | **72** | 74 |
| CTVAE-style | 34 | **95** | **97** | **89** | **96** | **97** | **91** | **97** | **98** | **96** | **97** | **98** | 52 | 68 | **83** |

(a) Style-consistency wrt. single styles of 3 classes (roughly uniform distributions).

| Model | 2 classes | | | 3 classes | | | 4 classes | | | 6 classes | | |
|---|---|---|---|---|---|---|---|---|---|---|---|---|
| CTVAE | 86 | 87 | 87 | 80 | 82 | 83 | **76** | 78 | 79 | 70 | 74 | 77 |
| CTVAE-info | 83 | 87 | 88 | 79 | 81 | 83 | 73 | 75 | 78 | 71 | 77 | 78 |
| CTVAE-mi | 86 | 88 | 88 | **80** | 81 | 84 | 71 | 74 | 79 | **73** | 76 | 78 |
| CTVAE-style | **97** | **98** | **99** | 68 | **97** | **98** | 35 | **89** | **95** | 67 | **84** | **93** |

(b) Style-consistency wrt. `DESTINATION(net)` with up to 6 classes (non-uniform distributions).

| Model | 2 classes | | | 3 classes | | | 4 classes | | | 6 classes | | | 8 classes | | |
|---|---|---|---|---|---|---|---|---|---|---|---|---|---|---|---|
| CTVAE | 91 | 92 | 93 | 79 | 83 | 84 | **76** | 79 | 79 | **68** | 70 | 72 | 64 | 66 | 69 |
| CTVAE-info | 90 | 90 | 92 | **83** | 83 | 85 | 75 | 76 | 77 | 68 | 70 | 72 | 60 | 63 | 67 |
| CTVAE-mi | 90 | 92 | 93 | 81 | 84 | 86 | 75 | 77 | 80 | 66 | 70 | 72 | 62 | 62 | 67 |
| CTVAE-style | **98** | **99** | **99** | 15 | **98** | **99** | 15 | **96** | **96** | 02 | **92** | **94** | **80** | **90** | **93** |

(c) Style-consistency wrt. `DISPLACEMENT` of up to 8 classes (roughly uniform distributions).

| Model | 2 styles | | | 3 styles | | | 4 styles | | | 5 styles | | |
|---|---|---|---|---|---|---|---|---|---|---|---|---|
| CTVAE | 67 | 71 | 73 | 58 | 58 | 62 | 49 | 50 | 52 | 27 | 37 | 35 |
| CTVAE-info | 68 | 69 | 70 | 54 | 58 | 59 | 48 | 51 | 54 | 28 | 32 | 35 |
| CTVAE-mi | 71 | 72 | 73 | 48 | 56 | 61 | 45 | 51 | 52 | 16 | 30 | 31 |
| CTVAE-style | **92** | **93** | **94** | **86** | **88** | **90** | **62** | **88** | **88** | **66** | **75** | **80** |

(d) Style-consistency wrt. multiple styles simultaneously.

Table 8: [min, median, max] style-consistency ($\times 10^{-2}$, 5 seeds) of policies evaluated with 4,000 basketball rollouts each. CTVAE-style policies significantly outperform baselines in all experiments and are calibrated at almost maximal style-consistency for 4/5 labeling functions. We note some rare failure cases with our approach, which we leave as a direction for improvement for future work.

| Model | Speed | | | Torso Height | | | B-Foot Height | | | F-Foot Height | | |
|---|---|---|---|---|---|---|---|---|---|---|---|---|
| CTVAE | 53 | 59 | 62 | 62 | 63 | 70 | 61 | 68 | 73 | 63 | 68 | 72 |
| CTVAE-info | 56 | 57 | 61 | 62 | 63 | 72 | 58 | 65 | 72 | 63 | 66 | 69 |
| CTVAE-mi | 53 | 60 | 62 | 62 | 65 | 70 | 60 | 65 | 70 | 66 | 70 | 73 |
| CTVAE-style | **68** | **79** | **81** | **79** | **80** | **84** | **77** | **80** | **88** | **74** | **77** | **80** |

(a) Style-consistency wrt. single styles of 2 classes (roughly uniform distributions).

| Model | 3 classes | | | 4 classes | | | | Model | 2 styles | | | 3 styles | | |
|---|---|---|---|---|---|---|---|---|---|---|---|---|---|---|
| CTVAE | 41 | 45 | 49 | 35 | 37 | 41 | | CTVAE | 39 | 41 | 43 | 25 | 28 | 29 |
| CTVAE-info | 47 | 49 | 52 | 36 | 39 | 42 | | CTVAE-info | 39 | 41 | 46 | 25 | 27 | 30 |
| CTVAE-mi | 47 | 48 | 53 | 36 | 37 | 38 | | CTVAE-mi | 34 | 40 | 48 | 27 | 28 | 31 |
| CTVAE-style | **59** | **59** | **65** | **42** | **51** | **60** | | CTVAE-style | **43** | **54** | **60** | **38** | **40** | **52** |

(b) Style-consistency wrt. `SPEED` with varying # of classes (non-uniform distributions).  (c) Style-consistency wrt. multiple styles simultaneously.

Table 9: [min, median, max] style-consistency ($\times 10^{-2}$, 5 seeds) of policies evaluated with 500 Cheetah rollouts each. CTVAE-style policies consistently outperform all baselines, but we note that there is still room for improvement (to reach 100% style-consistency).

| Model | Speed | Displacement | Destination | Direction | Curvature |
|---|---|---|---|---|---|
| CTVAE | 84 ± **1.0** | 72 ± **0.9** | 82 ± **0.6** | 77 ± 1.0 | 61 ± **0.8** |
| CTVAE-info | 85 ± 1.2 | 70 ± 1.2 | 81 ± 1.7 | 72 ± 1.2 | 60 ± 0.9 |
| CTVAE-mi | **86** ± 1.5 | 73 ± 1.5 | 82 ± 1.1 | 77 ± 1.1 | **71** ± 3.4 |
| CTVAE-style | 81 ± 31.4 | **94** ± 3.4 | **94** ± 3.8 | **97** ± 0.5 | 67 ± 12.6 |

(a) Style-consistency wrt. single styles of 3 classes (roughly uniform distributions).

| Model | 2 classes | 3 classes | 4 classes | 6 classes |
|---|---|---|---|---|
| CTVAE | 87 ± 0.7 | 82 ± **1.3** | **77** ± **1.7** | 75 ± **1.8** |
| CTVAE-info | 86 ± 1.9 | 81 ± 1.6 | 75 ± 2.9 | 76 ± 3.2 |
| CTVAE-mi | 88 ± **0.4** | 82 ± 1.8 | 75 ± 3.4 | 75 ± 2.0 |
| CTVAE-style | **98** ± 0.8 | **86** ± 14.4 | 74 ± 26.8 | **82** ± 13.0 |

(b) Style-consistency wrt. `DESTINATION(net)` with up to 6 classes (non-uniform distributions).

| Model | 2 classes | 3 classes | 4 classes | 6 classes | 8 classes |
|---|---|---|---|---|---|
| CTVAE | 92 ± 0.4 | 82 ± 2.6 | **78** ± 1.4 | **70** ± 1.4 | 66 ± **1.9** |
| CTVAE-info | 91 ± 0.8 | **84** ± **1.2** | 76 ± **0.6** | **70** ± **1.1** | 64 ± 3.2 |
| CTVAE-mi | 92 ± 1.4 | 83 ± 2.3 | 77 ± 2.5 | 68 ± 2.2 | 64 ± 2.5 |
| CTVAE-style | **99** ± **0.3** | 77 ± 41.2 | 75 ± 40.0 | 62 ± 42.9 | **88** ± 5.8 |

(c) Style-consistency wrt. `DISPLACEMENT` of up to 8 classes (roughly uniform distributions).

| Model | 2 styles | 3 styles | 4 styles | 5 styles |
|---|---|---|---|---|
| CTVAE | 70 ± 2.3 | 59 ± 1.7 | 50 ± 1.6 | 32 ± 3.1 |
| CTVAE-info | 69 ± 1.0 | 57 ± 2.3 | 50 ± 1.9 | 32 ± **1.7** |
| CTVAE-mi | 72 ± **0.8** | 52 ± 5.1 | 51 ± **0.8** | 26 ± 7.1 |
| CTVAE-style | **93** ± 1.2 | **88** ± 1.6 | **87** ± 2.5 | **76** ± 3.3 |

(d) Style-consistency wrt. multiple styles simultaneously.

Table 10: Mean and standard deviation style-consistency ($\times 10^{-2}$, 5 seeds) of policies evaluated with 4,000 basketball rollouts each. CTVAE-style policies generally outperform baselines. Lower mean style-consistency (and large standard deviation) for CTVAE-style is often due to failure cases, as can be seen from the minimum style-consistency values we report in Table 8. Understanding the causes of these failure cases and improving the algorithm's stability are possible directions for future work.

| Model | Speed | Torso Height | B-Foot Height | F-Foot Height |
|---|---|---|---|---|
| CTVAE | 57 ± 3.9 | 64 ± 3.1 | 67 ± 4.2 | 69 ± 3.7 |
| CTVAE-info | 58 ± **2.1** | 65 ± 4.2 | 64 ± 5.4 | 66 ± 2.7 |
| CTVAE-mi | 58 ± 3.9 | 66 ± 3.2 | 65 ± **3.6** | 70 ± 2.6 |
| CTVAE-style | **77** ± 5.3 | **81** ± **2.2** | **82** ± 5.4 | **77** ± **2.4** |

(a) Style-consistency wrt. single styles of 2 classes (roughly uniform distributions).

| Model | 3 classes | 4 classes |
|---|---|---|
| CTVAE | 45 ± 3.2 | 38 ± 2.9 |
| CTVAE-info | 49 ± **1.8** | 39 ± 2.8 |
| CTVAE-mi | 49 ± 2.2 | 37 ± **1.0** |
| CTVAE-style | **61** ± 2.9 | **51** ± 7.8 |

| Model | 2 styles | 3 styles |
|---|---|---|
| CTVAE | 41 ± **1.6** | 27 ± 1.9 |
| CTVAE-info | 42 ± 2.3 | 28 ± 2.2 |
| CTVAE-mi | 41 ± 4.9 | 29 ± **1.6** |
| CTVAE-style | **53** ± 6.1 | **43** ± 5.8 |

(b) Style-consistency wrt. `SPEED` with varying # of classes (non-uniform distributions).

(c) Style-consistency wrt. multiple styles simultaneously.

Table 11: Mean and standard deviation style-consistency ($\times 10^{-2}$, 5 seeds) of policies evaluated with 500 Cheetah rollouts each. CTVAE-style policies consistently outperform all baselines, but we note that there is still room for improvement (to reach 100% style-consistency).

| Model | Speed | | Torso Height | | B-Foot Height | | F-Foot Height | |
|---|---|---|---|---|---|---|---|---|
| | NLD | SC | NLD | SC | NLD | SC | NLD | SC |
| CTVAE-style | -0.28 | **79** | -0.24 | 80 | -0.16 | **80** | -0.22 | **77** |
| CTVAE-style+ | -0.49 | 70 | -0.42 | **83** | -0.36 | **80** | -0.42 | 74 |

Table 12: We report the median negative log-density per timestep (lower is better) and style-consistency (higher is better) of CTVAE-style policies for Cheetah (5 seeds). The first row corresponds to experiments in Tables 1 and 9a, and the second row corresponds to the same experiments with 50% more training iterations. The KL-divergence in the two sets of experiments are roughly the same. Although imitation quality improves, style-consistency can sometimes degrade (e.g. SPEED, FRONT-FOOT HEIGHT), indicating a possible trade-off between imitation quality and style-consistency.

| Model | Style-consistency ↑ | | | | | NLD ↓ |
|---|---|---|---|---|---|---|
| | Min | - | Median | - | Max | |
| RNN | 79 | 79 | 80 | 81 | 81 | -7.7 |
| RNN-style | 81 | 86 | 91 | 95 | 98 | -7.6 |
| CTVAE | 81 | 82 | 82 | 82 | 82 | **-8.0** |
| CTVAE-style | **91** | **92** | **97** | **98** | **98** | -7.8 |

Table 13: Comparing style-consistency ($\times 10^{-2}$) between RNN and CTVAE policy models for DESTINATION in basketball. The style-consistency for 5 seeds are listed in increasing order. Our algorithm improves style-consistency for both policy models at the cost of a slight degradation in imitation quality. In general, CTVAE performs better than RNN in terms of both style-consistency and imitation quality.

| | Speed | Displacement | Destination | Direction | Curvature |
|---|---|---|---|---|---|
| $\mathcal{L}^{\text{label}}$ | $3.96 \pm 0.33$ | $4.58 \pm 0.20$ | $1.61 \pm 0.18$ | $3.19 \pm 0.25$ | $28.31 \pm 0.95$ |

(a) Basketball labeling functions for experiments in section 6.1.

| | Speed | Torso Height | B-Foot Height | F-Foot Height |
|---|---|---|---|---|
| $\mathcal{L}^{\text{label}}$ | $3.24 \pm 0.83$ | $15.87 \pm 1.78$ | $17.25 \pm 0.73$ | $14.75 \pm 0.74$ |

(b) Cheetah labeling functions for experiments in section 6.1.

Table 14: Mean and standard deviation cross-entropy loss ($\mathcal{L}^{\text{label}}$, $\times 10^{-2}$) over 5 seeds of learned label approximators $C_{\psi^*}^{\lambda}$ on test trajectories after $n^{\text{label}}$ training iterations for experiments in section 6.1. $C_{\psi^*}^{\lambda}$ is only used during training; when computing style-consistency for our quantitative results, we use original labeling functions $\lambda$.

| | $M_\varphi$ test error |
|---|---|
| Basketball | $1.47 \pm 0.59 (\times 10^{-7})$ |
| Cheetah | $1.93 \pm 0.08 (\times 10^{-2})$ |

Table 15: Average mean-squared error of the dynamics model $M_\varphi$ per timestep per dimension on test trajectories after training for $n^{\text{dynamics}}$ iterations

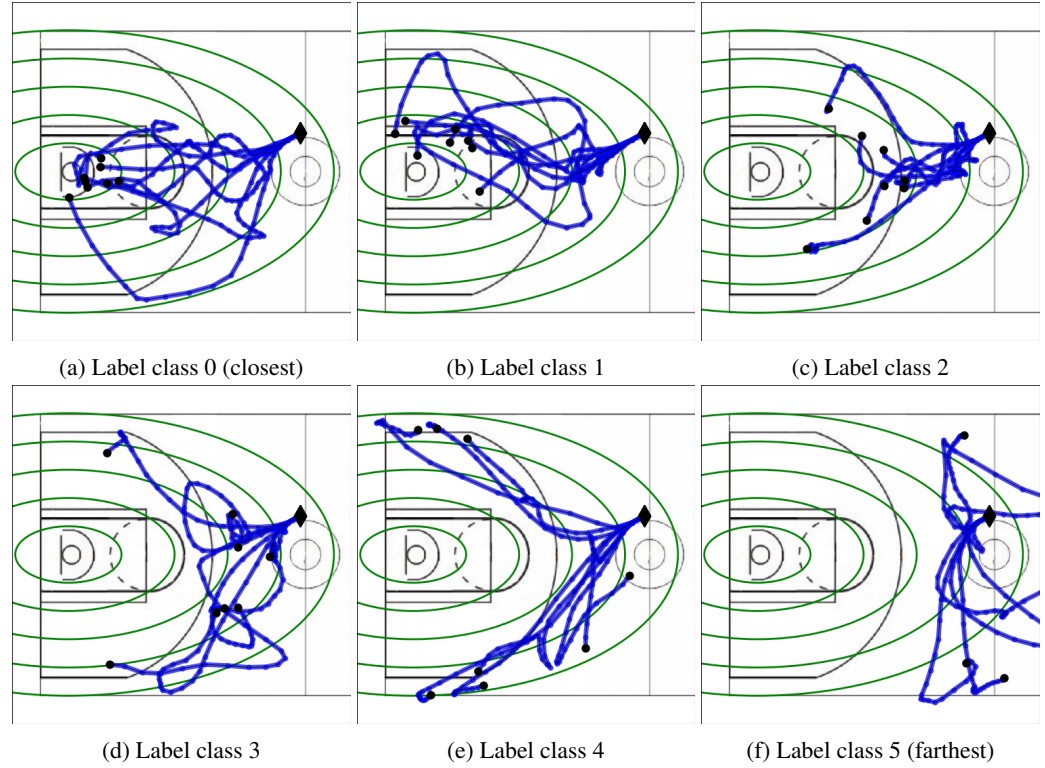

(a) Label class 0 (closest)    (b) Label class 1    (c) Label class 2

(d) Label class 3    (e) Label class 4    (f) Label class 5 (farthest)

Figure 4: Rollouts from our policy calibrated to `DESTINATION(net)` with 6 classes. The 5 green boundaries divide the court into 6 regions, each corresponding to a label class. The label indicates the target region of a trajectory's final position (●). This policy achieves a style-consistency of 0.93, as indicated in Table 8b. Note that the initial position (♦) is the same as in Figures 2 and 3 for comparison, but in general we sample an initial position from the prior $p(\mathbf{y})$ to compute style-consistency.

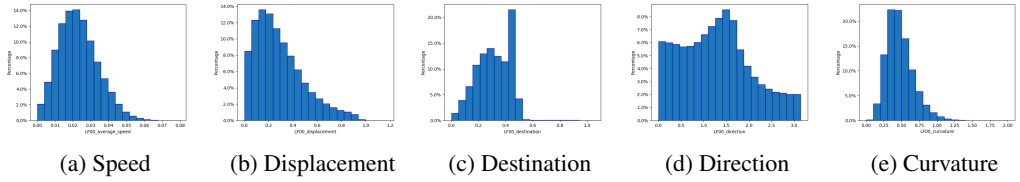

(a) Speed    (b) Displacement    (c) Destination    (d) Direction    (e) Curvature

Figure 5: Histogram of basketball labeling functions applied on the training set (before applying thresholds). Basketball trajectories are collected from tracking real players in the NBA.

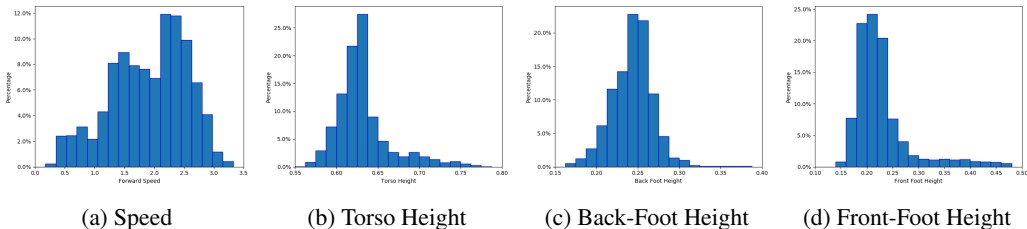

(a) Speed    (b) Torso Height    (c) Back-Foot Height    (d) Front-Foot Height

Figure 6: Histogram of Cheetah labeling functions applied on the training set (before applying thresholds). Note that SPEED is the most diverse behavior because we pre-trained the policies to achieve various speeds when collecting demonstrations, similar to (Wang et al., 2017). For more diversity with respect to other behaviors, we can also incorporate a target behavior as part of the reward when pre-training Cheetah policies.

