# OpenReview forum: "Learning Calibratable Policies using Programmatic Style-Consistency"
_ICLR.cc/2020/Conference — Reject_

### Official Review · AnonReviewer3 · 2019-10-14
**Official Blind Review #3**

**Rating:** 6

**Review:**

The authors propose learning generative models for long-term sequences that can take a style argument and generate trajectories in that style. The motivating example used is reconstructing expert trajectories from basketball games - trajectories can be sampled based on whether we want fast movement (SPEED), or whether they end close to the basket (DESTINATION).

It follows a data programming paradigm. We do not directly have style labels, so to get around this, we define labeling functions, which take a trajectory and output some boolean value. (Real valued labels are allowed but are not considered in this work). We assume the desired style is defined by some combination of labels, and that we know this combination (i.e. a fast trajectory to the basket should have the "speed above threshold c" label and "final location close to basket" label, which we have labeling functions for.)

Once we have this labeling function, we learn a trajectory VAE with a few loss functions. Standard behavioral cloning loss, and a style consistency loss that encourages labels of the generated trajectory to match labels of the target style. To make the optimization fully differentiable, we approximate non-differentiable labels with a learned labeling function (i.e. learn a classifier and then use classifier probabilities as the label), and learn a model of the environment to allow backprops through the rolled-out dynamics model for the entire trajectory. It's argued that learning a model helps with credit assignment, from an RL perspective credit assignment seems like the wrong word. Nothing about learning the model makes it easier to assign credit, the main gain it gives is making the problem differentiable.

On the basketball dataset, and a dataset of episode collected from the HalfCheetah MuJoCo environment, they demonstrate better style consistency. I had trouble finding an exact definition of style-consistency here - I assume it's defined as "given style c, how often does a trajectory sampled from pi(a|s,c) satisfy style c". It would be good to define this.

I would appreciate discussion on why style consistency classification is better than the mutual information baseline where MI between style labels and trajectory is maximized, it feels like they should be equivalent.

Overall I think this is a reasonable paper. The domains considered are fairly simple, but the idea seems sounds and the results seem good. I am concerned at all the requirements though - the method assumes the dynamics of the environment are learnable, and that we can define labeling functions that cover the space of styles we want, both of which seem like strong requirements.

**Experience Assessment:**

I do not know much about this area.

**Review Assessment: Checking Correctness Of Derivations And Theory:**

I assessed the sensibility of the derivations and theory.

**Review Assessment: Checking Correctness Of Experiments:**

I carefully checked the experiments.

**Review Assessment: Thoroughness In Paper Reading:**

I read the paper at least twice and used my best judgement in assessing the paper.

---

> ### Author Response · Authors · 2019-11-14
> **Response to Blind Review #3**
>
> > “... credit assignment seems like the wrong word.”
> Yes, a significant benefit of learning a dynamics model is that it allows us to differentiate through the environment dynamics. While this is not exactly credit assignment in the RL sense (e.g. we do not learn the value of each action with a Q-network), the problem being solved is similar in nature, in that the policy receives an informative learning signal for each action taken.
>
> > “exact definition of style-consistency”
> Style-consistency is defined in Equation (4) and your understanding is correct: trajectories sampled from policies calibrated to a style should be consistent with the style. We will make this clear in the text.
>
> > “why style-consistency classification is better than mutual information baseline.”
> One of the main insights in this work is that style-consistency is the true objective we wish to optimize for, and we introduce an algorithm to optimize for it directly. On the other hand, mutual information maximization optimizes style-consistency indirectly in Equation (13). There is no guarantee that the learned discriminative network r_psi in Equation (13) matches the true labeling function lambda, and this is reflected in our experiments (in fact, mutual information maximization is marginally better than the other baselines relative to our method).
>
> > “domains … are fairly simple”
> While one can, of course, work on much more challenging domains, we do think that our domains are already quite challenging.  Generating style-consistent trajectories over 200 time-steps is highly non-trivial.  We spent considerable effort in developing effective algorithmic approaches to address this “credit assignment” problem (there were many non-obvious design choices), and will release the software implementation upon publication.
>
> > “assumes the dynamics of the environment are learnable”
> Yes, we agree that this is a strong assumption for the algorithm we presented in the paper. However, possible extensions for future work is to consider model-free approaches, such as learning a style-conditioned value network to guide the policy during training. Another possible direction is to only enforce style-consistency at higher-level representations, similar to ideas presented in [1, 2].
>
> > “assumes … we can define labeling functions that cover the space of styles we want”
> We believe that this is a relatively weak assumption, as labeling functions are already prevalent in many real world applications. For example, sports analysts can define player roles based on positional heat maps [3]; aggressiveness in video games can be characterized by a player’s tendency to attack enemies [1]; and safety while driving can be quantified as the distance to the center of the lane [4]. There are many domains equipped with user-defined labeling functions that are naturally compatible with our framework.  Furthermore, our framework paves the way for semi-supervised approaches that can derive some of the labeling functions automatically.
>
> [1] Customizing Scripted Bots: Sample Efficient Imitation Learning for Human-like Behavior in Minecraft, Broll et. al.
> [2] Self-Consistent Trajectory Autoencoder: Hierarchical Reinforcement Learning with Trajectory Embeddings, Co-Reyes et. al.
> [3] Fine-Grained Retrieval of Sports Plays using Tree-Based Alignment of Trajectories, Sha et. al.
> [4] Batch Policy Learning under Constraints, Le et. al.

---

### Official Review · AnonReviewer1 · 2019-10-17
**Official Blind Review #1**

**Rating:** 3

**Review:**

This paper proposes a method to train style-conditional policies. The method has three components, a dynamics model, a labeling procedure and its approximation, and a policy. The model is trained on real trajectories, while the policy is trained on both real and simulated data coming from the model. The policy is trained to both imitate and be sensitive to the style labelling of states.
Such a method yields policies that can be executed with styles chosen externally, e.g. by a user.

This paper proposes a novel method that is an interesting take on imitation learning, but it is hard to judge how relevant this method is, as the paper has several inconsistencies and weaknesses that need to be resolved before it is accepted.

Inconsistencies: the reported NLL results do not match the trajectories shown in Figures. Some equations do not seem to reflect what is being optimized.
Weaknesses: Many quantities that should be reported are not: quality of models, diversity of policies, etc. The source of datasets should be clarified. (see detailed comments)

It's interesting that this method can leverage these very sparse or poor quality annotations, but it would be helpful to get a sense of how good the annotations provided here are. What's the threshold were annotation's quality is too low to be helpful? In the general case I suspect this threshold to be higher than what is hinted at in this paper, especially given Table 2. What happens in more realistic settings where e.g. users may specify hundreds of different labels?

Detailed comments:
- how dependent on having a variety of policies is this method? It's not clear how diverse the set of policies coming from the basketball dataset nor the Cheetah policies is, nor how this diversity affects learning. An experiment with explicitly different levels of diversity would strengthen understanding of this method. For Cheetah, it would be easy to report p(y) as a function of the target forward speed, that would give readers a sense of diversity for each label.
- For something like cheetah, with the labels that you propose being a very simple function of the state space, this somewhat resembles DIAYN [1] but with some grounding. Would it make sense to compare to a similar baseline?
- (Table 1) Why report (only) the median over 5 seeds? Papers usually report means. Plus, an indication of variance would be nice.
- It's not clear what the basketball dataset is. Where does it come from? Does it come with a simulator? If not, how do you evaluate style consistency or run step 11-12 of Alg. 2? Do you have a train/valid/test split to choose hyperparameters? (the appendix only suggests a train/test split)
- Why not cite MuJoCo? [2]
- For Figure 2 & 3, I suggest lowering the transparency/alpha value of the trajectories, as there is a lot of overlap.
- Table 1 is somewhat confusing. In (4) and the paragraph thereafter, you define \mathcal{L}^{style} as an error rate, i.e. when \lambda(\tau) \neq y, but in Table 1 you seem to report instead accuracy? (i.e. when \lambda(\tau) = y) If so, then you are reporting percentages, so \times 10^2 rather than 10^-2.
- You never report how well C_\psi is doing, and it's not clear to me why C_\psi is needed at all. When optimizing (8) are you directly treating (8)'s inner terms as negative rewards which is differentiated wrt \pi's parameters? If so, you are doing a form of DDPG, but there is a problem: after (4) you mention that L^style is not differentiable, meaning C_\psi doesn't provide gradient information to \pi. I assume that you acutally optimize (8) with L^label? Whether that's what you're doing or not, it should be clarified. If you are truly using L^style, then it's not clear why C_\psi is needed, as there is no differentiability anyways, and simply using \lambda directly will provide more signal.
- I'm somewhat perplexed by the values of Table 4. A log-likelihood of -190 represents a probability of 10^-83 (a _negative_ LL of -190 is, on the other hand, impossible by virtue of logs and probabilities, but I assume it is a "typo"). What is this the probability of? Entire trajectories? If so it would make more sense to report the _average_ log-likelihood, i.e. per timestep, because at this point in the paper, readers have no sense of how long trajectories are, and thus what these likelihoods represent. (for example, a NLL of 190 could be an average likelihood of 0.5 for 275 steps, or of 0.1 for 83 steps, or of .8 for 850 steps, which are all very different results! According to the appendix Table 5, the basketball trajectories are 25 steps long, which would mean that the imitation objective is not respected at all, exp(-190/25)=.0005, which would mean that pi(a|s) is .0005 on average.)
- Again Table 4, if the "-" is indeed a typo, then CTVAE-style is actually performing better than the baselines, especially for Cheetah (normally one wants negative log-likelihood to be as close to 0 as possible). Same for Table 10, if the NLL column is truly actually log-likelihood, then the "style+" objective really degrades imitation quality rather than improves it.

[1] Diversity is All You Need: Learning Skills without a Reward Function, Benjamin Eysenbach, Abhishek Gupta, Julian Ibarz, Sergey Levine
[2] MuJoCo: A physics engine for model-based control, Emanuel Todorov, Tom Erez Yuval Tassa

**Experience Assessment:**

I have published one or two papers in this area.

**Review Assessment: Checking Correctness Of Derivations And Theory:**

I assessed the sensibility of the derivations and theory.

**Review Assessment: Checking Correctness Of Experiments:**

I carefully checked the experiments.

**Review Assessment: Thoroughness In Paper Reading:**

I read the paper at least twice and used my best judgement in assessing the paper.

---

> ### Author Response · Authors · 2019-11-14
> **Response to Blind Review #1**
>
> > “Inconsistencies: … NLL results do not match the trajectories shown in Figures”
> The values we reported in Tables 4 and 12 are the average log-densities rather than the log-likelihoods. We apologize for the confusion and have clarified this in the text. See the discussion in the global comments for more details.
>
> > “Inconsistencies: … some equations do not reflect what is being optimized”
> Equation (8) indeed had a typo: we optimize with C_psi and L^label instead of L^style. This has been corrected in the text.
>
> > “Weaknesses: … quality of models”
> We’ve added Tables 14 and 15 in the appendix that report the test errors of our label approximators C_psi and dynamics models M_varphi respectively. The approximations are generally good enough for maximizing style-consistency (we hypothesize that better approximations will lead to better style-consistency, e.g. CURVATURE for basketball was harder to approximate and thus, CTVAE-style did not perform better than baselines). Note that label approximators C_psi are only used during training and not evaluation; we use the original labeling functions when computing style-consistency in our quantitative results.
>
> > “Weaknesses: … diversity of policies”
> We have included histograms of labeling functions for basketball and Cheetah in Figures 5 and 6 in the appendix to visualize the diversity of styles in training demonstrations. In our first set of experiments (Section 6.1) we threshold the labeling functions such that the labels are uniformly distributed. In our second set of experiments (Section 6.2) we apply thresholds at fixed intervals, which can lead to highly peaked and non-uniform distributions for some labeling functions.
>
> > “Weaknesses: … source of datasets should be clarified”
> The basketball dataset was collected from real NBA games, whereas we collected the Cheetah dataset ourselves by pre-training diverse policies (i.e. with slightly varying reward functions). The basketball dataset does not come with a simulator, but we assume a known dynamics function (next position = current position + velocity), which allows us to generate trajectories. See discussion about source of datasets in the global comments.
>
> > “What happens .. where users may specify hundreds of different labels?”
> This is a very challenging setting!  We believe our framework lays the groundwork for studying such settings, and would love to aspire towards that in the future.  For instance, a possible next step is to investigate how we can quickly calibrate to new styles without having to train a new policy from scratch; this can potentially have many connections with the original data programming paradigm, as well as multi-task learning.
>
> > “... resembles DIAYN [1] but with some grounding.”
> Yes, DIAYN [1] with grounding is similar to our CTVAE-mi baseline. DIAYN itself is a fully unsupervised algorithm in a reinforcement learning setting, but with no guarantee that a style a user cares about is represented among the skills that DIAYN learns. Our work focuses on the imitation learning setting, where we aim to calibrate to styles present in a collection of demonstration trajectories.  We can make this clearer in the text.
>
> > “Why report (only) the median over 5 seeds?”
> We expect that the practical use case of our framework is to run our algorithm over a few random seeds and then select the best one. From that perspective, we believe that the median best captures the reliability of our method in learning style-consistent policies. We also reported the min and max style-consistencies in Tables 8 and 9 of the appendix, which shows that our algorithm can sometimes have failure cases (in basketball). Understanding the cause of these failure cases and improving the stability of our algorithm are possible directions for future work. As per your request, we have included the mean and standard deviation style-consistencies as well in Tables 10 and 11 of the appendix. Note that low mean style-consistency (and high standard deviation) for CTVAE-style is exacerbated by the aforementioned failure cases.
>
> > “Do you have a train/valid/test split to choose hyperparameters?”
> We chose hyperparameters that appeared to work well for all baselines and kept them consistent for fair comparison. In a real application, we would perform a hyperparameter search to find the best training configuration.
>
> > “Table 1 … report … accuracy?”
> Yes, we report 1 - L^style, which can be interpreted as an accuracy. We mention this in the discussion of labeling functions used in our experiments.
>
> [1] Diversity is All You Need: Learning Skills without a Reward Function, Eysenbach et. al.

---

### Official Review · AnonReviewer2 · 2019-10-22
**Official Blind Review #2**

**Rating:** 3

**Review:**

The paper proposes a weak supervision method to obtain labels from functions that are easily programmable, and propose to use this for learning policies that can be "calibrated" for specific style. The paper demonstrates some experiments on a basketball environment and a halfcheetah environment, showing that the agent will perform according to corresponding styles.

My main concern here is the technical novelty of the proposed method: it seems that once we have the labels (which are limited to programmable functions), all we need to do is to learn a policy that conditions on the labels. In this case, we are not concerned with the latent variables whatsoever, therefore it seems that the CTVAE baselines are overkill for the task (learning latent variables that are not actually needed). Maybe more interesting baselines is to see how the two terms in (8) affect self-consistency performance, and not consider any methods that use unsupervised latent variables?

Minor questions:
	- The method's name, CTVAE-style is a bit confusing, since the policy does not depend on any latent variable z? At least from how the policy is described pi(\cdot |y) does not depend on unsupervised latent variables z.
	- Table 4, KL and NLL results do not seem to match? I wonder if the basketball kl should be multiplied by 10 and the cheetah ctval-style NLL is a typo?
        - Is it possible to extend this to continuous labels? This seems technically viable but unclear empirically.


**Experience Assessment:**

I have read many papers in this area.

**Review Assessment: Checking Correctness Of Derivations And Theory:**

I assessed the sensibility of the derivations and theory.

**Review Assessment: Checking Correctness Of Experiments:**

I carefully checked the experiments.

**Review Assessment: Thoroughness In Paper Reading:**

I read the paper at least twice and used my best judgement in assessing the paper.

---

> ### Author Response · Authors · 2019-11-14
> **Response to Blind Review #2**
>
> > “CTVAE-style does not depend on any latent variable?”
> We apologize for the confusing notation. All policies in our experiments are CTVAEs conditioned on both latent variables and style labels. We’ve updated the paper to make this clear in the experiments section.
>
> > “CTVAE baselines are overkill ...”
> Latent variable models generally improve imitation quality in sequential domains [1, 2, 3]. In particular, TVAE models have also been recently used in similar work for capturing diversity in behaviors [4, 5]. However, as also noted in our global comments, we emphasize that the choice of underlying policy model is orthogonal to our contributions, and we demonstrate this with an additional experiment where we train an RNN policy model instead. Table 13 in the appendix of the revised submission shows that even with a simpler RNN model, our algorithm still improves style-consistency.
>
> > “Table 4, ... results do not seem to match?”
> The values in Table 4 are correct. Note that NLD is the negative log-density instead of negative log-likelihood (see discussion in global comments). The NLD of CTVAE-style in Cheetah is also correct and indicates a tradeoff between imitation quality and style-consistency. We verify this in Table 12 when we show that more training iterations can improve imitation quality, but can sometimes degrade style-consistency.
>
> > “... continuous labels?”
> Yes, continuous labels also fall under our framework (e.g. by using mean-squared error for L^style). An interesting direction is to consider labeling functions that are already differentiable so we would not have to learn a differentiable approximator C_psi. We leave this for future work.
>
> [1] A Recurrent Latent Variable Model for Sequential Data, Chung et. al.
> [2] Sequential Neural Models with Stochastic Layers, Fraccaro et. al.
> [3] Z-Forcing: Training Stochastic Recurrent Neural Networks, Goyal et. al.
> [4] Robust Imitation of Diverse Behaviors, Wang et. al.
> [5] Self-Consistent Trajectory Autoencoder: Hierarchical Reinforcement Learning with Trajectory Embeddings, Co-Reyes et. al.

---

### Author Response · Authors · 2019-11-14
**Global comments to all reviewers**

We thank all reviewers for their insightful comments -- based on their feedback, we have revised the writing and added new experimental results that improve the paper in important ways. Our revised version has been uploaded. Here, we address common concerns (e.g. datasets, log-likelihood metrics), and we also respond to each review individually.

> Core Contributions

We first clarify our core contributions.

1) The first contribution is strategic: what systematic form of domain knowledge can we leverage to quickly and cleanly extract style information from raw behavioral data?  Our contribution is to leverage programmatic labeling functions crafted by domain experts.  Such labeling functions are readily available in many domains, and from that perspective, we view our work as opening new directions of research by studying how to formally and systematically leverage such information.  In that way, this contribution bears affinity to research that studies multimodal embeddings (e.g., how to systematically integrate image and audio).

2) The second contribution is formulaic: how can we formalize the learning objective to encourage learning style-calibratable policies? Our contribution is to formulate a new metric called style-consistency that captures and optimizes for the task objective: that trajectories sampled from a policy calibrated to a style should be consistent with that style.  Previous approaches use indirect methods for encouraging such notions of consistency, and we show that directly enforcing it can be very beneficial.  Furthermore, the underlying choice of the imitation learning algorithm is orthogonal to style-calibration, and we have included an additional experiment with a different underlying imitation learner (Appendix:Table 13) to verify this.

3) The third contribution is algorithmic: how do we design practical learning approaches that reliably optimize the learning objective?  Our contribution is an effective model-based approach for optimizing style-consistency over trajectories.  While the specific ingredients of our approach are each well-known, many of the design choices were non-obvious a priori, and we will release our implementation upon publication.

> Datasets

We chose 2 domains that have different problem characteristics --  Basketball does not have a simulator but plausibly admits a simple dynamics function (next position = current position + velocity); MuJoCo has an unknown dynamics function but lets us simulate trajectories. Furthermore, the Basketball dataset exhibits genuine real-world diversity in player behavior, while MuJoCo trajectories have less diverse agent behaviors (outlined below).

The basketball dataset was collected by STATS, a company that tracks real players from NBA games (~40 games in our case). Different players have different play-styles; for instance, a center or power forward may spend more time around the basket, while a faster player may cover more distance on the court. We describe these styles with DESTINATION and SPEED labeling functions respectively. We have added the distributions of styles returned by the labeling functions (Appendix:Figure 5) to visualize the diversity in the dataset.

For the MuJoCo Cheetah domain, we follow a similar procedure as [1] to collect diverse behaviors -- we pre-train Cheetah policies to walk at various speeds (the distribution of style labels is visualized in Appendix:Figure 6).

[1] Robust Imitation of Diverse Behaviors, Wang et. al.

> Log-likelihood

The values in Tables 4 and 12 are more accurately described as log-densities; we followed terminology from previous work in sequential generative models [2, 3, 4] that report this as “log-likelihoods”. Log-density of actions at each timestep is computed with respect to the parameters of a Guassian distribution output from our policy. Optimizing the CTVAE imitation learning objective will push the policy to assign higher density to actions observed in the training set.

As log-densities, the values in Tables 4 and 12 can be correctly interpreted. For instance, the average log-density per timestep in basketball is -7.9, which means the density is e^(-7.9) = 3.7e-4. The average variance output by the policy on test trajectories is 2e-5 per dimension. Thus, the average mean-squared error is roughly 2.16e-5 per dimension, which indicates that the imitation learning objective is indeed respected.

We apologize for the confusion. Tables 4 and 12 now scale log-densities per timestep, and we will update the text to clarify this.

[2] A Recurrent Latent Variable Model for Sequential Data, Chung et. al.
[3] Sequential Neural Models with Stochastic Layers, Fraccaro et. al.
[4] Z-Forcing: Training Stochastic Recurrent Neural Networks, Goyal et. al.

---

### Decision · Program_Chairs · 2019-12-19

**Decision:**

Reject

**Comment:**

The reviewers generally reached a consensus that the work is not quite ready for acceptance in its current form. The central concerns were about the potentially limited novelty of the method, and the fact that it was not quite clear how good the annotations needed to be (or how robust the method would be to imperfect annotations). This, combined with an evaluation scenario that is non-standard and requires some guesswork to understand its difficulty, leaves one with the impression that it is not quite clear from the experiments whether the method really works well. I would recommend for the authors to improve the evaluation in the next submission.